# A Novel Method to Determine the Optimal Location for a Cellular Tower by Using LiDAR Data

**Shruti Bharadwaj** [1,†] , **Rakesh Dubey** [1,†] , **Md Iltaf Zafar** [1,†] , **Saurabh Kr Tiwary** [1] , **Rashid Aziz Faridi** [2] **and Susham Biswas** [1,*]

1    Department of Computer Science and Engineering, Rajiv Gandhi Institute of Petroleum Technology, Amethi 229304, India; pgi17001@rgipt.ac.in (S.B.); pgi19001@rgipt.ac.in (R.D.); pgi15001@rgipt.ac.in (M.I.Z.); eche17048@rgipt.ac.in (S.K.T.)
2    Department of Geography, Aligarh Muslim University, Aligarh 202001, India; rashid.faridi@gmail.com
*    Correspondence: susham@rgipt.ac.in; Tel.: +91-915352704626
†    These authors contributed equally to this work.

**Abstract:** The cellular industry faces challenges in controlling the quality of signals for all users, given its meteoric growth in the last few years. The service providers are required to place cellular towers at the optimal location for providing a strong cellular network in a particular region. However, due to buildings, roads, open spaces, etc., of varying topography in 3D (obstructing the signals) and varying densities of settlements, finding the optimal location for the tower becomes challenging. Further, in a bigger area, it is required to determine the optimum number and locations for setting up cellular towers to ensure improved quality. The determination of optimum solutions requires a signal strength prediction model that needs to integrate terrain data, information of cellular tower with users' locations, along with tower signal strengths for predictions. Existing modeling practices face limitations in terms of the usage of 2D data, rough terrain inputs, and the inability to provide detailed shapefiles to GIS. The estimation of optimum distribution of cellular towers necessitates the determination of a model for the prediction of signal strength at users' locations accurately. Better modeling is only possible with detailed and precise data in 3D. Considering the above needs, a LIDAR data-based cellular tower distribution modeling is attempted in this article. The locations chosen for this research are RGIPT, UP (45 Acre), and Shahganj, Agra, UP, India (6 km²). LiDAR data and google images for the project sites were classified as buildings and features. The edges of overground objects were extracted and used to determine the routes for transmission of a signal from the tower to user locations. The terrain parameters and transmission losses for every route are determined to model the signal strength for a user's location. The ground strength of signals is measured over 1000 points in 3D at project sites to compare with modeled signal strengths (an RMSE error 3.45). The accurate model is then used to determine the optimum number and locations of cellular towers for each site. Modeled optimum solutions are compared with existing tower locations to estimate % over design or under design and the scope of improvement (80% users below −80 dB m improves to 70% users above −75 dB m).

**Keywords:** cell phones; transmission tower; LiDAR; GIS; signal strength; optimum location

## 1. Introduction

A cell phone tower is an important component of the network since it transmits signals to cell phones, and its strength varies with the location of the tower. The strength of a cell phone tower's signal is measured in decibel milliwatts (dB m). The dead zone occurs when a cell phone user (receiver) has an excellent reception of the network signal at one location but low reception on the other surrounding side at the same time. The cell phone signal has a relatively low power output (typically at less than 1 milliwatt). The signal strength is typically measured in the negative dB m range in broad areas. In India, a signal

strength of $-50$ dB m is considered to be excellent for a given location. When an urban area must give acceptable signal strength to every area, a mobile phone tower must be placed in an optimal location and at an appropriate height. In addition, the optimal site is determined by the placements of the building's features, terrain data points, ground points, and other things in the vicinity [1]. As a result, the transmission of the cell tower signal is determined by the placements of the towers and the routes that the signal takes before hitting the end of any user location (receiving point). Starting from the source, a signal might have a direct or indirect path, i.e., from the position of a cell phone tower before arriving at any user location (receiver). In the indirect path, the signal can propagate via diffraction (top of building and around the sides) or reflection (from the ground or via wall). Signal propagation involves signal attenuation owing to distance, diffraction, reflection, and other factors [2,3]. The selection of an appropriate location necessitated the use of 3D terrain data, which is often represented as point cloud data with x, y, and z values. It comes in the form of a raster, vector, or point cloud and presents a challenge to extract the terrain's features [4–6].

Fulfilling the demands of an increasing number of users requires cell phone towers to be placed at the optimal location, and it is a challenging task [7]. For achieving accurate tower placement, several research works have been performed in the cell placement field [8]. In research by Deane et al. [9], an attempt to explain a potent algorithm to identify the optimal cell phone tower positions by approaching three algorithms was done. Among these three, the first one is the greedy algorithm (by making the best possible choice that seems best at the particular moment), the second happens by the ratio of heuristics, and the third one by the intelligent genetic algorithm [10]. It was concluded that the genetic one is the best long-term algorithm. Another study by AL-Hamami et al. [11] uses a geographical information system and a spatial data mining system to find the best possible tower location. This research was accomplished by using the Digital Elevation Model (DEM) on the satellite project area image [12–14].

Similarly, a three-step algorithm was proposed by Kashyap et al. [15] for finding out the best height of the cell phone tower. The algorithm computes the most cost-effective position and altitude for tower placement to achieve suitable signal strength [16–18]. To maximize the cell tower placement, geospatial mining with a Geographic Information System (GIS) is used as a tool in research by AL-Hamami et al. [11]. A review regarding signal monitoring is discussed by the author in [19,20].

In study [21], the geographic data from multiple sources, comprising satellite data, topographical maps, local digital maps, and specifications of existing towers [22] (such as latitude and longitude, antenna height, and frequency bands), in the project region (see Figure 1) was investigated. Using these data, ArcGIS software was used to create multiple layers to find the best location. Dead zones in cell transmissions and areas of an existing cellular tower that overlap have been identified [23]. Hence, the study area's existing towers were analyzed to determine the best locations for tower placement [24]. These prior studies use a GIS environment to evaluate [25] the need for acceptable signal strength by locating many cell phone towers, but they do not focus on the best position for cellular phone towers or how the signal can cover every receiver position while retaining optimal strength [26].

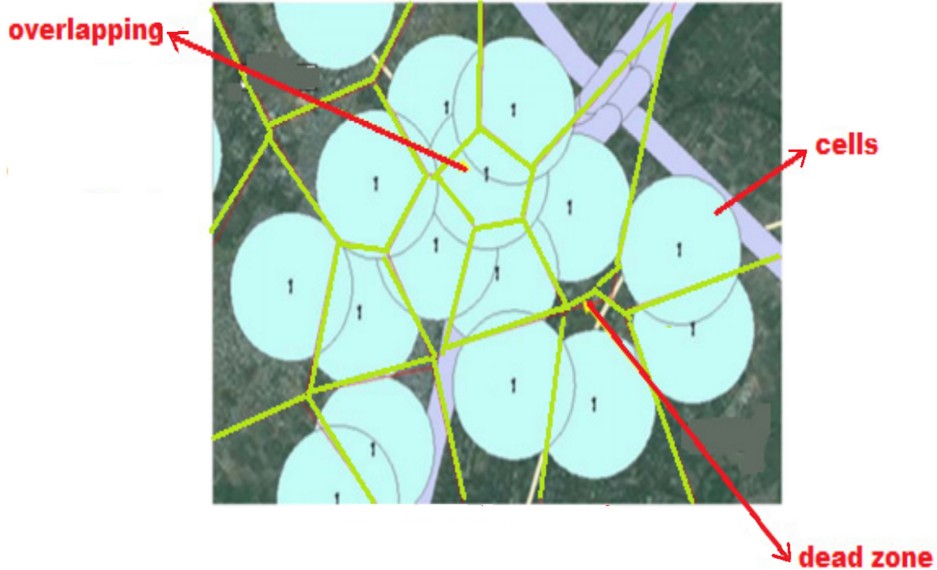

**Figure 1.** Reduction of weak area and interference (Reprinted from Ref. [21]).

To understand how the signal reaches the user's location, the initial step is to understand the propagation of the signal [27–29]. Cell phones use radio waves to communicate. In this study, it has been made clear that this signal transmission follows two paths: one is the line of sight (direct path), and the other is non-line of sight (obstructed by buildings, barriers, etc.) [30,31], as shown in Figure 2a. In means of propagation, it is analog to sound transmission. While propagating from cell phone tower to cell phone, user signal suffers different losses [32]. These losses are due to fading and multiple propagations [33]. From the studies, it has been concluded that the range of the excellent signal is between 0 and −60 dB m and for the poor signal is above −100 dB m [34], as shown in Figure 2b. The goal of the suggested research is to figure out where the best cell phone tower placement is. Users play a vital part in determining path loss and attenuation in the strength of the signal, thus efforts have been made here to evaluate them and then utilize them to determine the ideal transmission route from cell phone towers to cell phones users to ascertain the optimal cell phone tower location [35].

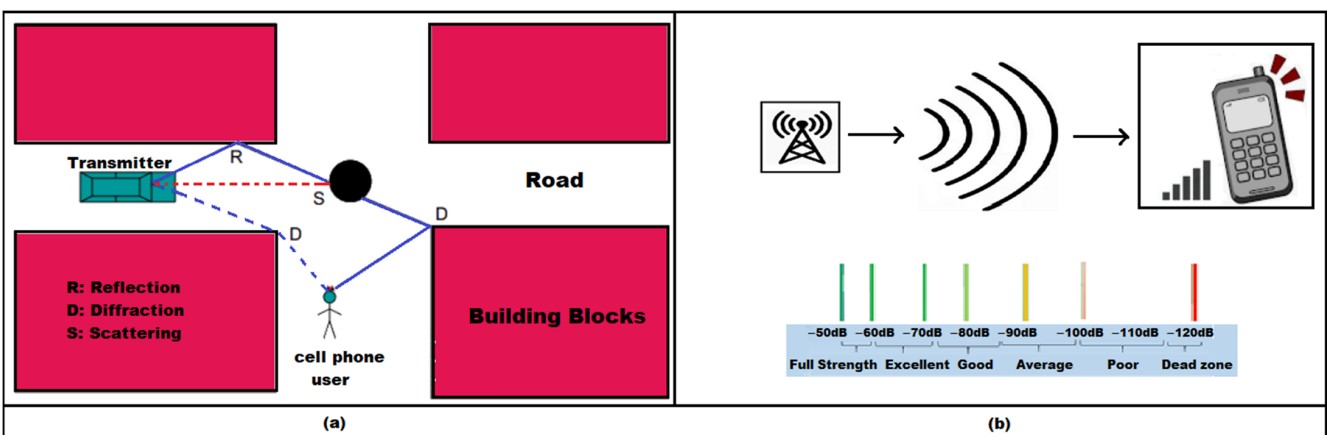

**Figure 2.** (**a**) Signal propagation, (**b**) Signal strength range.

## 2. Research Question on the Placement of Cellular Tower(s)

Increased usage of cell phones and the need for higher bandwidth necessitates placing cellular towers in optimal locations. However, there are several issues associated with placing a cellular tower in a particular place. In the GIS environment, primarily the spatial

and non-spatial attributes are considered to determine the optimal solution for finding out the cell tower location(s). The distance of towers from cities, highways, other towers is weighed considering their geographic location along with non-spatial attributes of population, income, etc., for the determination of optimal location. The determination of all these distances requires the creation of shapefiles for cities, highways, populations, etc. Shapefiles are also required to be in the same coordinate system to carry out any spatial analysis, e.g., the position of a cellular tower near the city, highway, city, highway. Spatial analysis in GIS is often carried out through the generation of buffer areas around each shapefile based on pre-decided distance and comparing them by overlay analysis. Thus, the buffer areas around cities, highways, and existing towers are required to be determined. The buffer raster analysis indicates the locations where there is no coverage and overlapping areas. Further various weighting criteria are used for the distance to the city, highway, or other towers to decide an optimum location for setting up a new tower.

It is challenging to find the optimal location of setting up a cellular tower for an area having land cover objects of different sizes, shapes, and volumes and having land uses of different types. Similarly, finding the best locations for setting up multiple cellular towers to cover a city is difficult. In computer science, it can lead to solving an NP-Hard problem.

GIS-based models for cellular distribution do not consider the cases of the presence of multiple towers in an area of different service providers. It can so happen that the quality of signal in a place may be better from a service provider compared to another. The model should be able to distinguish it and suggest a better service provider for a place. GIS-based modeling for cellular towers demands significant data processing. It is required to produce the shapefiles for city, highway, tower, population, income rate, detailed land cover, and land uses. Further, these shapefiles are required to be compared with the location of the tower(s) to determine the spatial objective function giving appropriate weights in line with importance. Different models suggested by researchers are either theoretical solutions or GIS-based solutions indicating sufficiency of distribution or dead zones. Primarily, the research does not verify their prediction with the ground data. Thus, the efficiencies of these solutions are doubted.

The existing optimum cellular tower locating algorithms are primarily theoretical or are using very simplified rough 2D terrain data of land-use and landcover to ascertain the best location for setting up a cellular tower using simplified objective functions. In the absence of a detailed and practical approach to adapt to any urban setup precisely, the GIS approaches follow average techniques to determine the optimality. It invariably results in over or under design. There is, thus, a need to evolve a technique that should be able to use the detailed variation in topography for an area and estimate the detailed obstructions or impacts of each for the signals to reach any user location from a cell phone tower. It is required to be understood that the urban environment, having adjacent high-rise buildings located between the cellular tower and user location, plays a very big role in terms of the obstruction of signals. Thus, 2D data and rough land use and landcover data significantly preclude the estimation of detail and accurate signal strength for a place. Developing countries have a large proportion of unplanned cities, offering a very mixed land use and land cover and often creates a significant challenge in terms of the accurate estimation of signal strength for a place from a cellular service provider. Thus, it is required to come up with a technique that can model or compute accurate signal strength for a user location from a cell phone tower. The model should also be able to estimate the best location for setting up a cellular tower for a place or estimate numbers and locations of possible cellular towers for an area to provide good signal strength to every user location. The existing GIS software suffers in terms of its ability to accommodate huge terrain data or extract buildings and other obstructions in 3D automatically, which prevents direct transmission of a signal from the cellular tower(s) to different user locations. There is thus a requirement to use 3D terrain data of high precision, which can find out a technique to extract all obstructions between any pair of the cellular tower and user locations, determine losses in transmission at its every route of transmission, and can estimate an optimization algorithm that can work

with all the spatial attributes (transmission paths, distances, transmission losses, etc.) and non-spatial attributes (number of users at a building, population density, etc.) to determine the signal strength and optimum location(s) for the setting-up of cellular tower(s). LiDAR data offers very precise 3D data in point cloud form. It is required to establish a novel cellular tower locating optimization technique using LiDAR data, which will overcome challenges of accommodating precise 3D data and enable highly accurate optimization in terms of offering the best location(s) for the setting-up of cellular tower(s) and giving the best estimate of signal strength for any surrounding areas.

### 3. Methodology

The determination of an accurate model for the estimation of the optimum location and number of cellular towers involved the estimation of a technique consisting of the following components:

(a) Establishment of an accurate model to predict the signal strength at a user location after transmission of a signal from a cellular tower location.

    a.    Development of a model inputting accurate 3D terrain data, cellular tower and user location(s), and signal strength at tower location to determine the strength at desired user locations.

    b.    Extraction of detailed routes for transmission of signals using 3D LiDAR data.

    c.    Estimation of terrain parameters and dependent transmission losses for every route between each tower and user pair as a modeling input.

    d.    Verification of the model for a wide variety of terrain geometry and a large number of points.

    e.    Generation of 3D signal strength map for a cellular tower to its all-neighboring users' locations.

(b) Establishment of a technique to determine the optimum location for setting up a cellular tower using an accurate signal strength prediction model.

    a.    Estimation of optimum location in X, Y, and Z for an area.

    b.    Determination of optimality at the rough grid in X, Y, e.g., 100 m × 100 m, then refinement to a very fine grid, 2 m × 2 m.

    c.    Determination of optimality in height (Z) about heights of user's locations.

(c) Establishment of a technique to determine the optimum number and locations of the cellular tower for an area to offer good/acceptable quality of signal strength to most users.

    a.    Estimation of number and locations of the cellular tower using a signal strength prediction model and optimality prediction model.

    b.    Establishment of global optimality maintaining a trade-off between over design and under design and good/adequate signal strength requirement for an area.

The need for establishing a technique to determine the optimum location(s) for the setting up of cellular towers accurately was understood. The technique would depend on the use of very accurate 3D terrain data, which in the present study was used with LiDAR data. It is required to determine all the routes through which the signal can transmit from a cellular tower location to a user location. The routes can be direct where there is no obstruction between the tower and user location; it can also be indirect where there is/are obstruction(s) between the two. The route determination algorithm required the extraction of buildings, roads, open spaces, etc., between the cellular tower and the users' locations. Once routes can be determined, these are used to determine terrain parameters (e.g., path difference, path length, reflecting ground) for the computation of signal losses. The information of locations of the tower (or the potential location of the tower) and a user pair, topographical obstructions between them, and the strength of signal of the cellular tower can be integrated using a model to predict the likely signal strength for the user location. The model derived can be used for predictions in *one-to-one* or *one-to-many* modes. The prediction of signal strength whenever applied for one cellular

tower and one location for user comprised the *one-to-one* mode, while when there were many users to one cellular tower location, it constituted the *one-to-many* prediction mode. Essentially *one-to-many* is used for making a map of predicted signal strength for a tower location. The model integrating signal strength of tower, locations for tower and users, and the transmission losses (dependent on terrain parameters) are designed in line with the transmission mechanisms of radio waves (analogous with the transmission of the sound wave). The signal strength prediction model's semi-empirical equations were established with the least square technique. The modeling equations took ground observations with a signal strength of 500 points around an existing location of the tower to comprehensively validate the model. The signal strength prediction model is used next to determine the optimum location in (X, Y) and in Z to set up the cellular tower. The optimum location (X, Y) for the setting up of a new tower primarily worked around the centroid location for the area for a grid size of 100 m × 100 m. Initially, areas of different size and shapes were tested for centroid centric optimality and are fine-tuned to establish the optimal location at finer grids (5 m × 5 m, 2 m × 2 m) near the rough solution for the precise 3D urban setup. Targeting potential solutions for the refinement of the solution is planned to reduce computational time for optimality to a great extent. The optimum height for the optimal location of the tower is planned by testing various heights beginning at near ground heights to heights above the tallest building for the area. Two sites are considered for the testing and establishment of a comprehensive model for the prediction of signal strength at users' locations. These are at RGIPT campus, UP, India (45 Acres in the rural backdrop) and Agra-Shahganj, UP, India, 6 km$^2$ in an urban city. Following the establishment of the prediction model, it is planned to estimate the optimum location for setting up the tower(s) for the RGIPT and Agra sites. The number of cellular towers required to cover the bigger Agra site is modeled. The existing tower locations and their numbers are compared with modeled numbers and locations for the area to find out the extent of under or over designs for cellular towers. The detailed methodology adapted is schematically explained below.

A system is designed and developed for finding the best position for installing a cell phone tower to achieve sufficient signal strength in each area. This gives an adequate signal strength at the user's location. The methodology is designed in a way that consists of two models. One is the cell phone signal strength determination model, and the other one is the model for the determination of optimal location(s) for setting up a cell phone tower(s). The methodology is explained by the image shown in Figure 3. Determination of the optimal location of cell phone tower(s) needs a cell phone determination model.

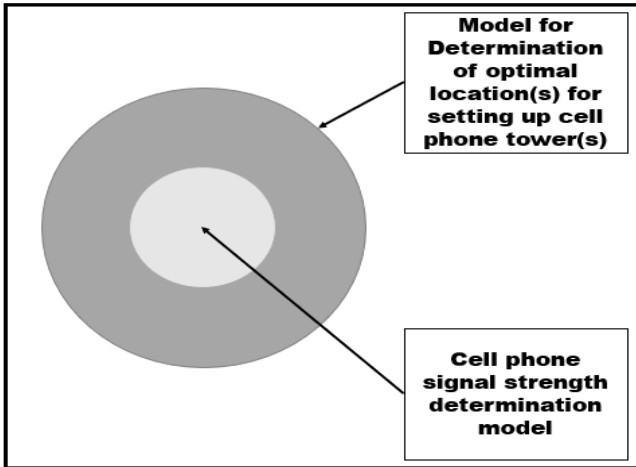

**Figure 3.** Proposed methodology.

The cell phone signal strength determination model is a model that is proposed to determine the signal strength of cell phone users. Cell phone signals transmit similarly to noise or sound signals, as shown in Figure 4.

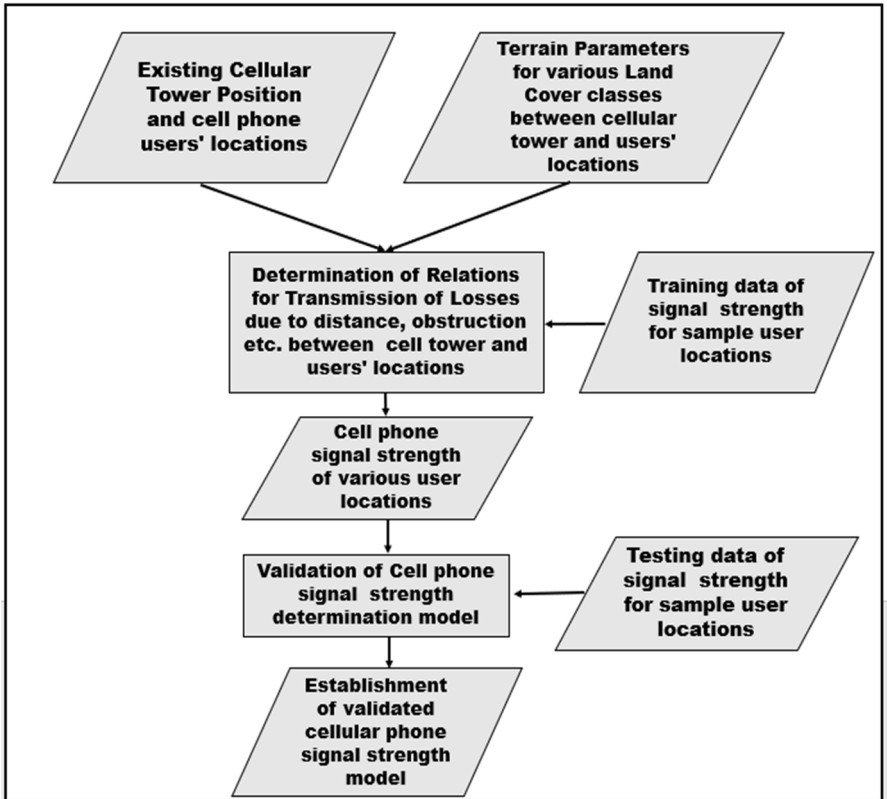

**Figure 4.** Cell phone signal strength determination model.

Starting with the collection of existing cellular tower locations for the RGIPT campus, India, we stored the cell phone signal strength at various user's locations through the mobile app (Network cell info lite).

The terrain parameters between the cellular tower and the user's location are determined by an analogous property between cell phone signal propagation with sound propagation model.

The terrain determination used the LiDAR dataset, which is a 3D point cloud (X, Y, Z) that includes direct information on topographical elements, such as buildings, vegetation, and ground. As a result, data processing necessitates the extraction of terrain points that block and/or govern signal transmission. The terrain parameters determination methodology is shown in Figure 5.

If there is no obstacle between the cell phone tower and user location(receiver), the signal moves directly between them. However, if there is/are barrier(s) between them, the signal travels indirectly by diffraction, reflection, and other means [26]. As a result, measuring the signal strength at each user location (receiver) necessitates calculating the transmission path(s).

LiDAR data has been collected for the RGIPT campus project area. The pathway of a signal must be identified using clear information about the building's corners or edges, as well as any other impediments that may exist between the cell phone tower and the receiver (i.e., user location). Edges and corners of buildings or obstacles that may occur in the pathway between the cell phone tower and the cell user's location are extracted [36]. The pathways are calculated using a plane cutting approach.

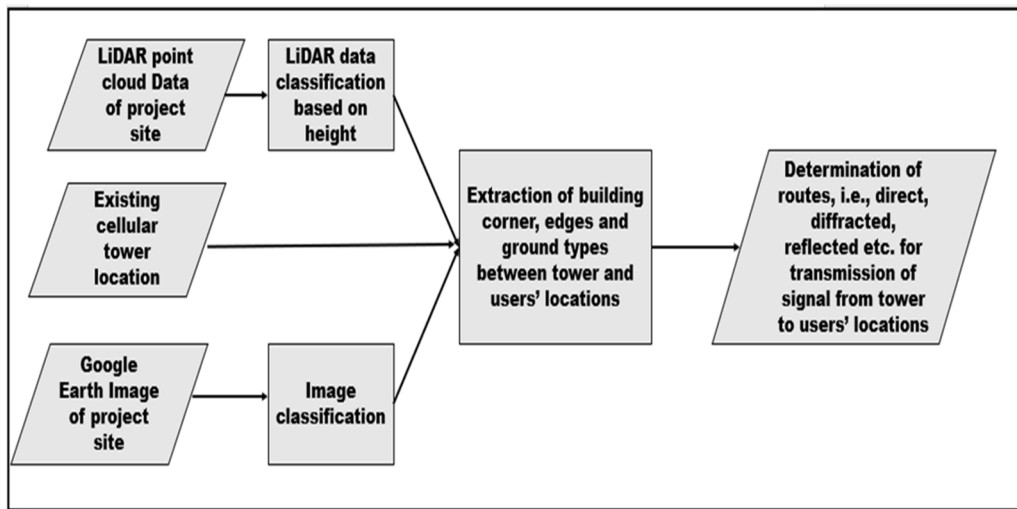

**Figure 5.** Terrain parameter determination between tower and users' locations.

The methodology entails establishing an algorithm for extracting all-terrain features/terrain data points from the LiDAR dataset without changing the point data of DEM [37]. After determining the specific signal pathways, an equation is fitted for a model to calculate the signal strength and signal attenuations (signal strength losses).

For an equation fitting by the least square method, we collected 1500 points throughout the RGIPT campus. For these 1500 points, 1000 points are chosen as the training set, and 500 points are set as the testing set. The non-linear equation will be fitted because the signal propagation follows the sound propagation model analogy and depends non-linearly on distance.

After getting the determined relationship between signal strength and terrain parameters. The validation is performed with the 500 testing points after the establishment of a validated signal strength model is performed.

Two areas that have been chosen are a rural area and an urban area for the determination of the optimal locations of the cell phone tower. The rural area (that is, the RGIPT campus) contains fewer buildings, more vegetation, and fewer cell phone towers. The urban area (the Shahganj area of Agra city) where there are more buildings, less vegetation, and more cell phone towers. Two different areas are taken to fit a model with an equation that can sustain terrain feature variation.

Cell phone tower locations are proposed for the RGIPT campus by the method from step 1 to step 6, as shown in Figure 6. The Urban area (Shahganj area of Agra city) contained the five previously installed towers. For an adequate signal strength of a location, five proposed locations of towers are calculated by the method from step 7 to step 13, as shown in Figure 6.

To calculate signal strengths (or relative transmission loss) at diverse ground locations, an algorithm is developed to work efficiently for each pair of cell phone towers and its neighboring user location. The study aims to look at signal strength (or relative signal strength loss) for different user locations (receiver) [38].

Once the boundaries of the building and other obstacles are identified for a certain topographical condition with locations of building blocks, ground, trees (vegetation), etc., the model will endeavour to estimate the best pathway for signal transmission, the relative loss of signals at different receiver/user locations, and the best placement (X, Y, and Z) coordinates for a cell phone tower. The optimal location for a cellular tower will guarantee that the signal strength is sufficient for all nearby cell phone users (receivers).

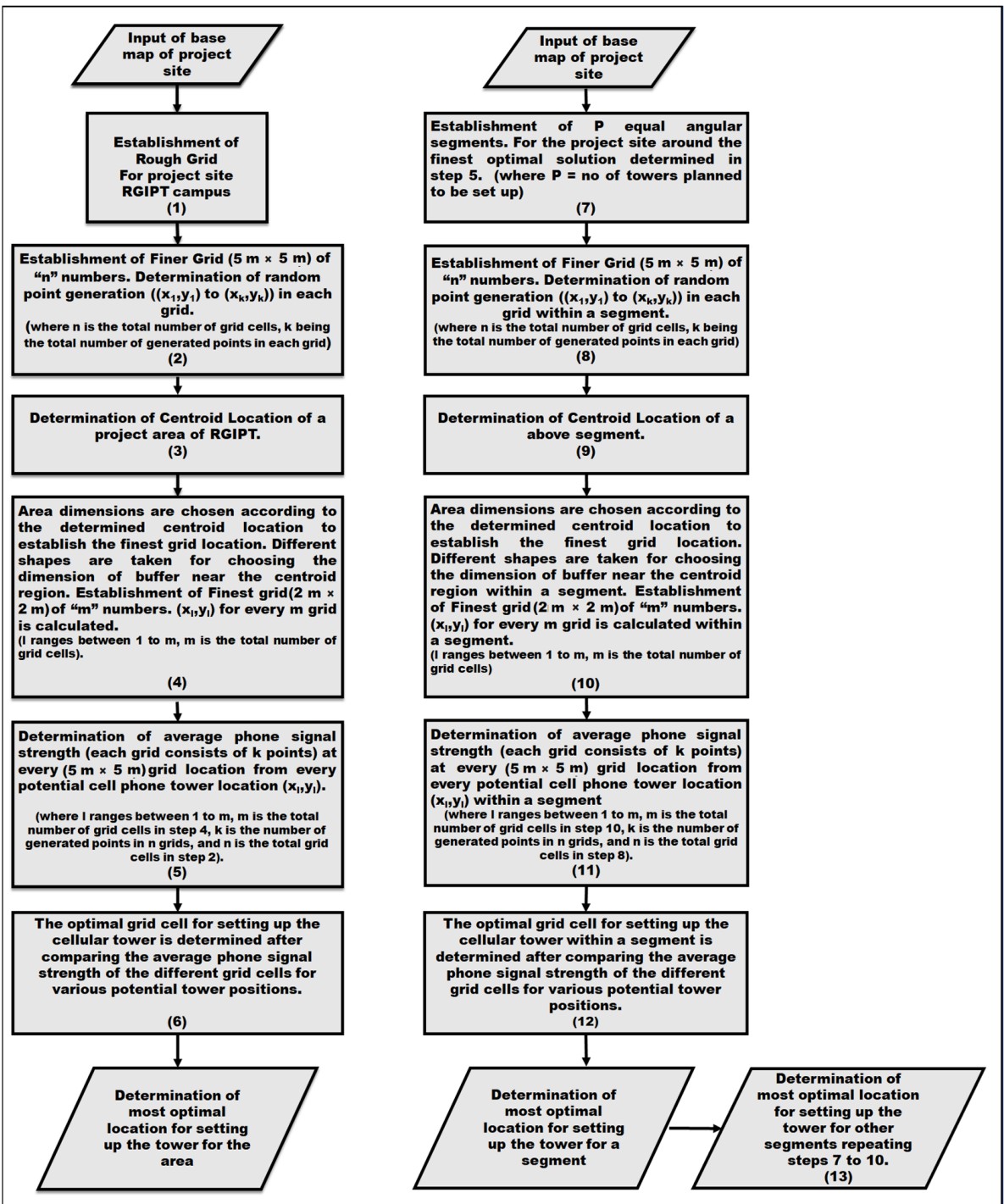

**Figure 6.** Model for the determination of optimal location(s) for setting up a cell phone tower(s).

Cell phone tower location is also determined for the case where the population ratio is non-uniform. For that case, population density is considered, which area needs a higher value of signal strength. In some cases, where the population is non-uniform and the optimal location is found that is providing moderate signal strength to the high population part as well as low population part, then a shift in the cell phone tower location to improve the signal strength for the high population part will be desired.

## 4. Results and Discussions

For finding the optimal position of the Cell phone tower, the following steps are mentioned in the resulting flow. This result flow states the result of each step. The process splits into two steps: One is the "Cell Phone Signal Strength Determination Model", the second is the "Model for Determination of optimal location(s) for setting up cell phone tower(s)". These two steps give results as discussed in the result flowchart, shown in Figure 7.

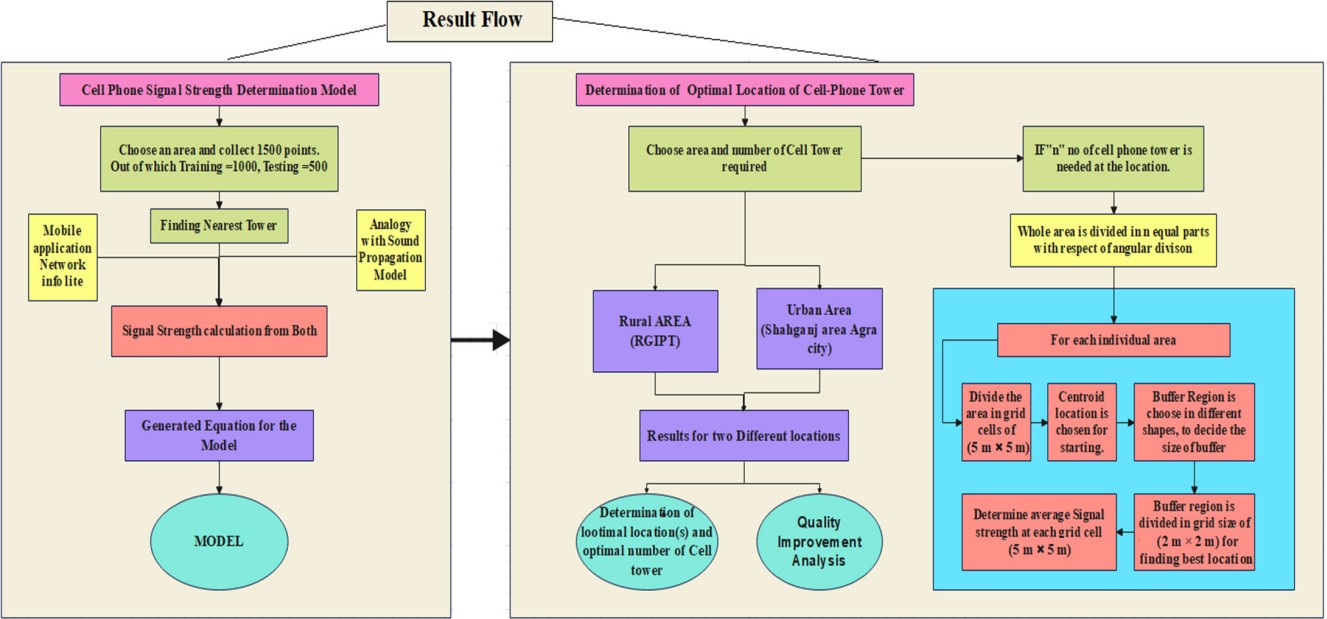

**Figure 7.** Result Flowchart.

### 4.1. Determination of the Strength of the Signal

The model represents the collection of signal strength from the existing cell tower location. During transmission, the signal follows direct or indirect paths from the cell tower to the user's location. This model is going to determine a relationship for transmission of losses due to distance, obstruction, etc., between the cell tower and users' locations.

4.1.1. Location of Existing Towers and Cell phone Users

For the study, a project area of the RGIPT campus was taken, which lies in between (latitude (26.265788), longitude (81.504372)) and (latitude (26.263355), longitude (81.515723)). Adjoining the campus, there is a railway line.

The project area of the RGIPT campus is surrounded by the Jio and Voda towers, as shown in Figure 8. Except for the Jio tower, the rest of the towers are far from the RGIPT location. The main aim of this research is to find an optimal location for a cell phone tower. This optimal location will provide adequate signal strength to the user's location. The cell phone will catch the signal strength from the cell phone tower that is near to the location. The nearest tower will give better signal strength.

Although there are two types of towers near the RGIPT location, one is a Jio tower and the other is a Vodafone tower. The closest to the RGIPT location is "Jio tower A". "Jio tower A" is at location (latitude 26.263130°, longitude 81.515706°). "Jio tower A" (nearest tower) is considered for the further processing of deploying the tower for the RGIPT location. "Jio tower A" plays an important role in the cellular signal strength of RGIPT.

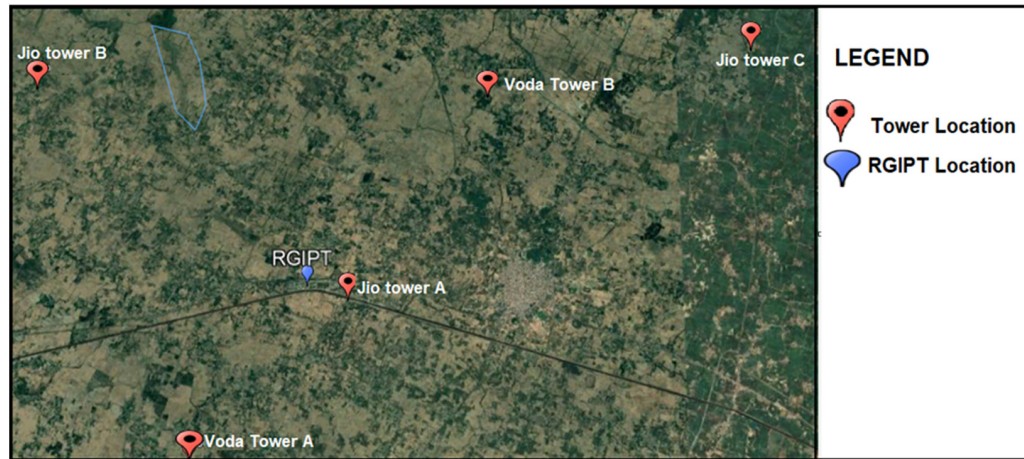

**Figure 8.** Tower locations near the RGIPT campus.

From the project area, the 1500 points were taken as a user's location. At these points, signal strength due to the nearest cell phone tower was recorded by a mobile application named (Network cell info lite), shown in Figure 9. The cell phone tower is not limited to only one network. It can be for multiple networks.

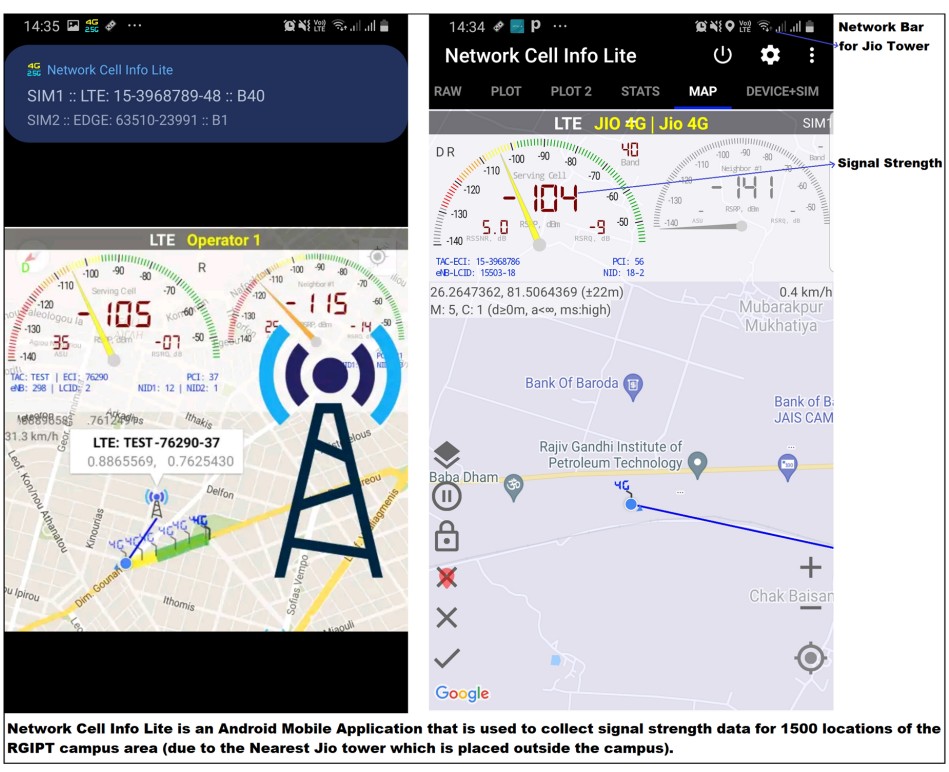

**Figure 9.** Mobile app for signal strength collection, Network cell info lite.

An existing signal strength map from Jio tower A for the RGIPT campus made in ArcGIS is shown in Figure 10b. This signal strength map is calculated for 1500 locations of the RGIPT campus.

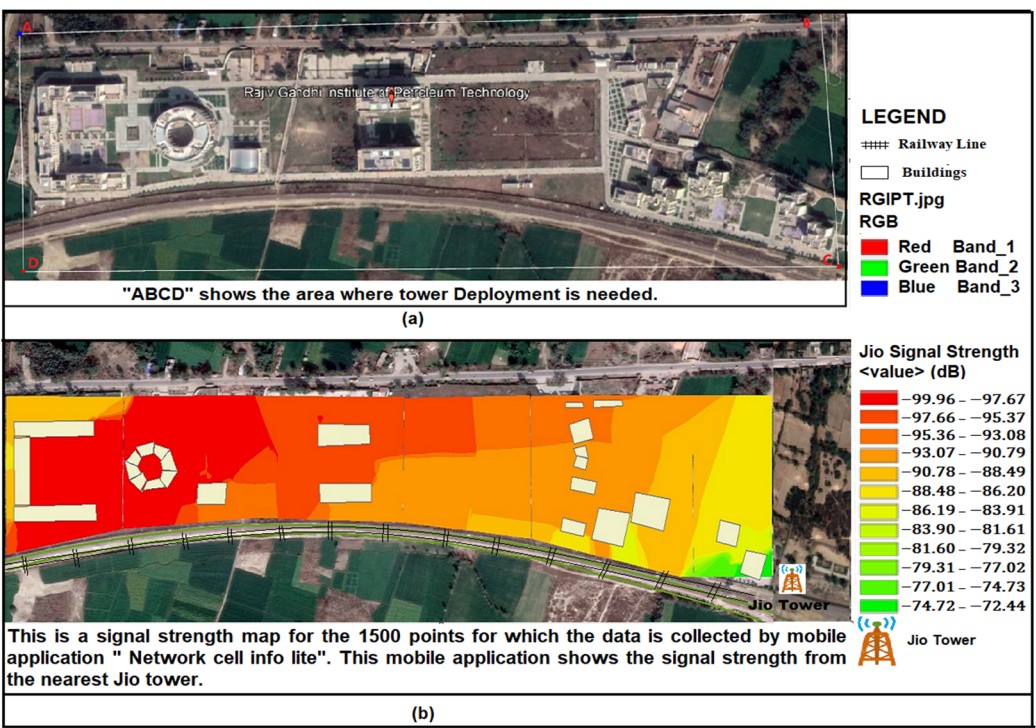

**Figure 10.** (**a**) Project Area of RGIPT campus, (**b**) Signal strength from nearest JIO cell phone tower at 1500 user locations on the RGIPT campus.

4.1.2. Terrain Parameter Determination

Signal propagation of cell phone towers is analogous to a noise/sound propagation model. The determination of terrain parameters depends on signal transmission.

Signal transmission pattern: Signals from cell phone towers suffer losses when traveling from one location to another, as proven by our own experiences. These losses are due to distance and barriers present between the cell phone tower and the cell phone user. While talking about attenuation losses in signal strength, there are some analogies between the losses in signal transmission and losses in sound transmission. They both have similar transmission patterns. They primarily travel directly from the source to the destination (in the case where there are no obstacles between them). When there is/are an obstacle(s) between them, it can also move indirectly by diffraction, reflection, and other means. The propagation path of the signal must be defined to determine the signal strength. The transmission path of the signal is calculated by evaluating the direct path or indirect path from the cell phone tower to the cell phone user location [2]. These paths (path over the top of building a barrier, path around the sides of the building, and path after reflection from ground and wall of a building) help to determine the parameters, such as distance and barrier attenuation. The transmission of the path is calculated by the path determination of the signal, as shown in Figure 11 below.

Determination of signal strength equation: This step determines the equation for finding the signal strength at any location. The equation is fitted analogously to the semi-empirical formulae used in sound propagation modeling.

$$Sound\ level\ (Receiver) = Sound\ level\ (source) - Distance\ attenuation - B \cdot A \qquad (1)$$

$$Distance\ Attenuation = 20\ log_{10}(D) + 11 \qquad (2)$$

$$B \cdot A = 5.65\ + 66N + 244N^2 + 287N^3 \qquad (3)$$

$$N = \frac{Path\ difference}{\frac{\lambda}{2}} \qquad (4)$$

$$\lambda = \frac{c}{f} \tag{5}$$

$$Path\ Difference = D1 - D \tag{6}$$

where *D* = direct transmission path (m)

*D1* = indirect transmission path (m)

*B·A* = barrier attenuation (dB)

λ = wavelength (m)

*c* = speed of light (m/s)

*f* = frequency (Hz)

*N* = Fresnel number

For the RGIPT area, LIDAR point cloud data is generated [39], as shown in Figure S1a. From the LIDAR data, the digital elevation model is created, which is shown in Figure S2b. The LIDAR data and the DEM consist of the hostel area, academic building (AB1 and AB2), and teacher's housing of the RGIPT campus. Signal transmission paths were obtained from the information of building corners and edges estimated in [40]. This will determine the distance attenuation and barrier attenuation.

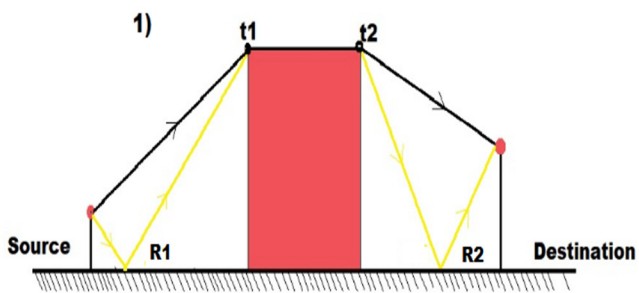

**P1(path 1)= Source – t1 – t2 – Destination**
**P2(path 2)= Source – R1 – t1 – t2 – Destination**
**P3(path 3)= Source – t1 – t2 – R2 – Destination**
**P4(path 4)= Source – R1 – t1 – t2 – Destination**

In this figure t1,t2 are top intersection point on building edge and R1,R2 are ground reflection point between Source and t1, between t2 and Destination.

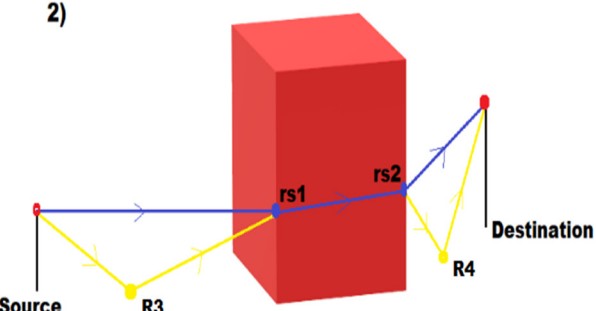

**P5(path 5)= Source – rs1 – rs2 – Destination**
**P6(path 6)= Source – R3 – rs1 – rs2 – Destination**
**P7(path 7)= Source – rs1 – rs2 – Destination**
**P8(path 8)= Source – R3 – rs1 – rs2 – R4 – Destination**

In this figure rs1,rs2 are right side intersection point on building edge and R3,R4 are ground reflection point between Source and rs1, between rs2 and Destination.

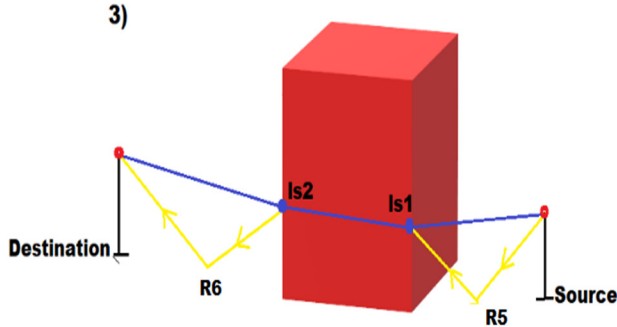

**P9 (path 9)=Source – ls1 – ls2 – Destination**
**P10(path 10)=Source – R5 – ls1 – ls2 – Destination**
**P11(path 11)=Source – ls1 – ls2 – R6 – Destination**
**P12(path 12)= Source – R5 – ls1 – ls2 – R6 – Destination**

In this figure ls1,ls2 are left side intersection point on left side of building edges and R5,R6 are ground reflection point between source and ls1, between ls2 and destination.

**Figure 11.** Principal path of propagation explaining diffraction and reflection.

### 4.1.3. Relation between Signal Strength and Terrain Parameters

For equation fitting, an area of the RGIPT campus has been taken. Signal strength data of 1500 points are observed by "Network cell info lite".

According to the study, distance attenuation is a factor of the logarithm of distance, and the barrier attenuation is dependent on path difference; that is, the difference between the direct path and indirect path between the source and receiver. A signal is calculated in terms of dBm and at the cell phone tower location, the signal strength is considered to be 0 (dB m). "*SR*" is denoted for signal received.

$SR_{Calculated}$ (user location) = Signal (cell phone tower) − Attenuation losses

As per the analogy with propagation models, the attenuation losses are a function of distance and barrier attenuations.

$$Attenuation\ Losses = (a \times log_{10}d) + B{\cdot}A \tag{7}$$

Signal (cell phone tower) = 0 dB m

$$SR_{Calculated} = -((a \times log_{10}d) + B{\cdot}A) \tag{8}$$

$$SR_{Observed} = -((a \times log_{10}d) + B{\cdot}A + Error) \tag{9}$$

Training data is 1000 points out of 1500 points. A total of 500 points are for testing purposes. For each point, $SR_{Observed}$ (dB m) is captured from "Network cell info lite".

For point 1, the equation is

$$SR_{Observed1} = -((a \times log_{10}(d_1)) + B{\cdot}A_1 + Error) \tag{10}$$

For point 2, the equation is

$$SR_{Observed2} = -((a \times log_{10}(d_2)) + B{\cdot}A_2 + Error) \tag{11}$$

For point 3, the equation is

$$SR_{Observed3} = -((a \times log_{10}(d_3)) + B{\cdot}A_3 + Error) \tag{12}$$

and so on, for point 1000, the equation is

$$SR_{Observed1000} = -((a \times log_{10}(d_{1000})) + B{\cdot}A_{1000} + Error) \tag{13}$$

where $d_1$, $d_2$, $d_3$ ... $d_{1000}$ are the direct distance in meters from the source tower to the receiver location. $B{\cdot}A_1, B{\cdot}A_2, B{\cdot}A_3 \ldots , B{\cdot}A_{1000}$ are the barrier attenuation (dB) that occurred due to buildings in between the source tower and receiver.

We can write the above equations as

$$Y = MX + C \tag{14}$$

where Y = *SR* Observed (user location) + $B{\cdot}A$

$$X = -log_{10}(d) \tag{15}$$

$$C = -Error \tag{16}$$

Y, X are known values, and *M, C* are unknowns.

All the equations from 10 to 13 are now written as:

For point 1,

$$Y_1 = MX_1 + C \tag{17}$$

For point 2,

$$Y_2 = MX_2 + C \tag{18}$$

For point 3,

$$Y_3 = MX_3 + C \tag{19}$$

$\dots$

For point 1000,

$$Y_{1000} = MX_{1000} + C \tag{20}$$

To find the best-fit line, we try to solve the above equations (from 17 to 20) with the unknowns $M$ and $C$. As the 1000 points do not lie on a line, there is no actual solution, so instead, we compute a least-squares solution putting our linear equation into matrix form. Computing:

$$A \cdot x = b \tag{21}$$

$$A = \begin{pmatrix} X_1 & 1 \\ X_2 & 1 \\ X_3 & 1 \\ \vdots & \vdots \\ X_{1000} & 1 \end{pmatrix}, \ B = \begin{pmatrix} Y_1 \\ Y_2 \\ Y_3 \\ \vdots \\ Y_{1000} \end{pmatrix}, \text{ and } x = \begin{pmatrix} M \\ C \end{pmatrix} A^T = \begin{pmatrix} X_1 & X_2 & X_3 & \cdots & X_{1000} \\ 1 & 1 & 1 & \cdots & 1 \end{pmatrix} \tag{22}$$

where $A^T$ is the transpose of $A$.

Multiplying $A^T$ to both sides of Equation (21), the equation becomes:

$$A^T A \cdot x = A^T \cdot b \tag{23}$$

$$\begin{pmatrix} X_1 & X_2 & X_3 & \cdots & X_{1000} \\ 1 & 1 & 1 & \cdots & 1 \end{pmatrix} \times \begin{pmatrix} X_1 & 1 \\ X_2 & 1 \\ X_3 & 1 \\ \vdots & \vdots \\ X_{1000} & 1 \end{pmatrix} \times \begin{pmatrix} M \\ C \end{pmatrix} = \begin{pmatrix} X_1 & X_2 & X_3 & \cdots & X_{1000} \\ 1 & 1 & 1 & \cdots & 1 \end{pmatrix} \times \begin{pmatrix} Y_1 \\ Y_2 \\ Y_3 \\ \vdots \\ Y_{1000} \end{pmatrix} \tag{24}$$

$$\begin{pmatrix} \sum_{i=1}^{1000} X_i{}^2 & \sum_{i=1}^{1000} X_i \\ \sum_{i=1}^{1000} X_i & \sum_{i=1}^{1000} 1 \end{pmatrix} \times \begin{pmatrix} M \\ C \end{pmatrix} = \begin{pmatrix} \sum_{i=1}^{1000} X_i \times Y_i \\ \sum_{i=1}^{1000} Y_i \end{pmatrix} \tag{25}$$

$$\begin{pmatrix} \left( \sum_{i=1}^{1000} X_i{}^2 \right) \times M + \left( \sum_{i=1}^{1000} X_i \right) \times C \\ \left( \sum_{i=1}^{1000} X_i \right) \times M + \left( \sum_{i=1}^{1000} 1 \right) \times C \end{pmatrix} = \begin{pmatrix} \sum_{i=1}^{1000} X_i \times Y_i \\ \sum_{i=1}^{1000} Y_i \end{pmatrix} \tag{26}$$

The final equations from the ($2 \times 2$) matrix in Equation (26) are:

$$\sum_{i=1}^{1000} X_i \times Y_i = \left( \sum_{i=1}^{1000} X_i{}^2 \right) \times M + \left( \sum_{i=1}^{1000} X_i \right) \times C \tag{27}$$

$$\sum_{i=1}^{1000} Y_i = \left( \sum_{i=1}^{1000} X_i \right) \times M + \left( \sum_{i=1}^{1000} 1 \right) \times C \tag{28}$$

After solving Equations (27) and (28), the value of $M$ = 18.11 and $C$ = 34.89.

$$Signal\ Strength\ Equation = -18.11 \times log_{10}d - B \cdot A - 34.89 \tag{29}$$

4.1.4. Validation

For the validation, the fitted equation is being tested for the 500 testing points. For these testing points, the RMSE error is 5.48. While comparing a list of values of 500 points

observed and calculated signal strength, 25 points have outlier values, which means values deflect more from the observed value.

The RMSE error for the estimated equation is 3.45 after removing the outliers from the data.

Why use this equation? The equation is taken as a relationship of logarithmic distance for signal strength calculation while talking about the linear relation between distance and signal strength. Y = $a$X + $b$, where X is distance, a and b are variable.

$$b = B{\cdot}A + Error$$

The linear fitted equation has the RMSE of 77.15.

$$Signal\ Strength\ Equation = -0.4 \times d - B{\cdot}A - 20 \qquad (30)$$

This equation signifies that moving towards the increasing distance of cell tower and cell phone users, there is a big change in the value of signal strength.

The signal strength map after calculating signal strength at user locations by using the equation is shown in Figure 12.

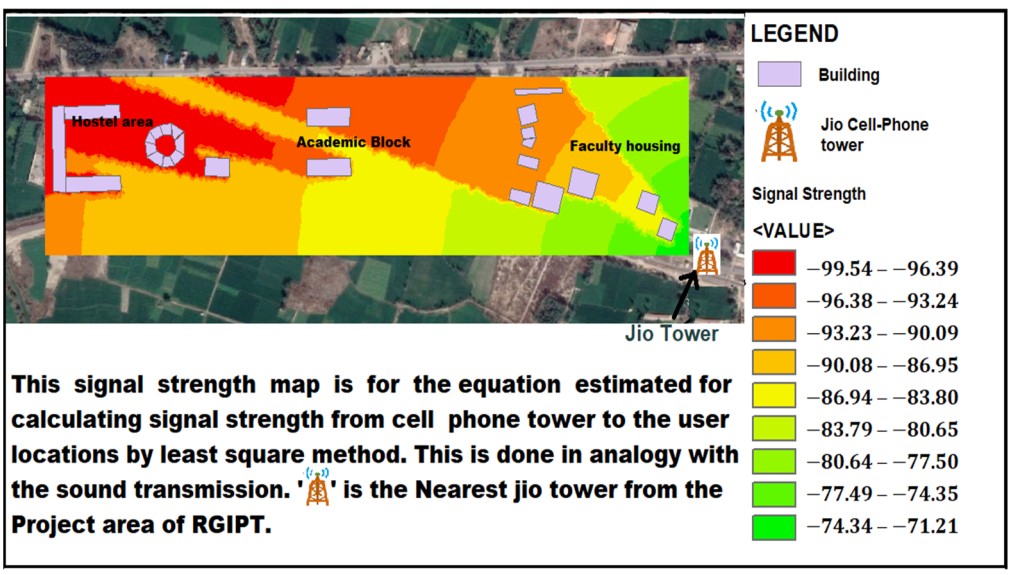

**Figure 12.** Signal Strength after the determination of the signal strength equation from the nearest JIO cell phone tower on the RGIPT campus.

*4.2. Optimal Location(s) for Setting up Cell Phone Tower(s)*

To determine the optimal location, we have chosen a two-way procedure, one is to a rural area (that is, the RGIPT campus where there is only one JIO tower nearby) and the second one is to an urban city area (that is Shahganj areas of Agra city where there are many JIO towers nearby). The two different areas are chosen as they have different terrain parameters, such as soft ground, hard ground, crowded area, vegetation.

4.2.1. Rural Area (RGIPT Campus)

The area of the RGIPT campus is in a rural area where there is a lot of vegetation, fewer buildings, and only one Jio cell phone tower outside the campus. Now, after getting the equation for calculating signal strength, there is a need to estimate the X and Y coordinate values that define the position and the Z coordinate value [41] that defines the height of the cellular tower for the placement of a cell phone tower in the project area of the RGIPT campus containing buildings. The estimation of X, Y, and Z coordinates gives the optimal location of a cellular tower to manage adequate signal strength in a large area.

The determination of the Z coordinate of cell phone tower: The Z parameter is required to be determined for finding the optimal height of the cell phone tower. For estimating the Z value, X and Y are fixed at the center of an area, and Z keeps on varying [4]. The Z parameter is determined in reference, which signifies that the height of a cell phone tower is set equal to or more than the height of the highest building in the area. For determining the Z of cell phone tower, the tower is initially kept at an X and Y centroid of that area and then the height of the tower starting from 0, 5, 15, 20, 30, and 35 m is taken one by one (described in the supplementary file as Figures S3–S5).

Determination of the X and Y coordinates of the cell phone tower: After getting the height estimation of the tower, the final optimum location X and Y of the tower are determined by the following steps given below:

Step 1: Taking the required project area of RGIPT where the deployment of the tower is processing, which is shown in Figure 10. Then, we have to inscribe the area in a rectangle named "ABCD", as shown in Figure S6a. After this, there is a need to determine the building's edges and corners, which is discussed in [4]. The determined buildings are then plotted in MATLAB over the project area of the RGIPT campus, as shown in Figure S6b.

Step 2: For the placement of a cell phone tower, there is a need for a space from (2 m × 2 m) to (5 m × 5 m). Therefore, here we make partitions of the area in several blocks (5 m × 5 m), shown in Figure S7a.

For the optimal location, there is a need to have some random user locations to check the signal strength. Random points are generated for each block that is found in step 2. For these random points, signal strength will be calculated, and the log sum of these values gives the final value of signal strength to that block. Now the question is, how can we choose the number of random points to be generated. Therefore, we check the final signal strength value of block for 3 points, 5 points, 7 points, 10 points, and come up with the results that if we have 4 corners of a block it is the least accurate to choose points that are less than 4. For the rest, there is little difference in the final signal strength value when we talk about 5, 7, and 10, as shown in Figure S7b. Therefore, we choose 5 points for a block.

Step 3: For the deployment of a cell tower in a project area. The area fitted in a rectangle "R" (ABCD) is now ready to locate one cell phone tower at the centroid of the rectangle "R", as shown in Figure S8. "C" is the centroid of rectangle "R". The calculated "C" is not accurate for the uneven area, so we choose a square taking "C" as Centroid. The dimensions of the square taken here are decided on the basis of different types of areas mentioned.

Step 4: For the deployment of a cell tower in a project area. Different shapes are taken for choosing the area dimension of the buffer near the centroid region. The establishment of the finest grid (2 m × 2 m) of "$m_g$" numbers, (xl, yl) for every $m_g$ grid is calculated (l ranges between 1 to $m_g$, $m_g$ being the total number of grid cells).

L-shaped area (100 m × 100 m)—We fit the rectangle in an irregular-shaped area. Find out the centroid and the distance of the centroid from the outer wall of the irregular area. For the initial case, the dimension is taken as 30 m × 30 m, and we divide the area into blocks of dimensions (2 m × 2 m) as a cell phone tower requires this much area to establish. Now for each block, there is a centroid named c1, c2, c3 . . . cn, as shown in Figure S9.

Now the whole area is L-shaped, which is now divided into several blocks as mentioned earlier, i.e., 5 m × 5 m dimension. Five random points are generated in each block of the L-shaped area. Centroids c1, c2 . . . cn will be considered as cell phone tower locations one by one and the average signal strength of each block of L-shaped area will be checked.

A threshold is set for each block, checking signal strength; if the signal strength lies between −50 and −70 dB m, then the count will increase by 1, while the rest remains the same. The centroid tower location is c1, c2, c3 . . . cn, and whichever gets the maximum count will be the perfect location for the cell phone tower. Next, we check the distance between the centroid "C" and the new tower location out from the c1, c2, c3 . . . cn. Then, from Figure S10 shown below, the new centroid location is determined as C'.

L-shaped area (50 m × 75 m)—Repeat the above procedure and find out the distance between the centroid "C" and the new tower location. The distance is 8.5 m, as shown in Figure S11a.

U-shaped area (one tail short, one long)—Similarly, the above procedure is repeated for the case again, and the distance is 7 m, as shown in Figure S11b.

After analyzing different shaped cases and calculated distances, it is clear that the dimension of the square created at centroid "C" should be taken as half of the distance of the centroid to the outlines of the area. The length is mentioned as "L".

Step 5: Determination of average phone signal strength (each grid consists of five random points) at every 5 m × 5 m grid location from every potential tower location (xl, yl). Where l ranges between 1 and m, m being the total number of grid cells in step 4, and n is the total number of grid cells in step 2.

Step 6: Signal strength map for cell tower located at the finest tower location. From the tower location to five random points in each block after dividing the project area into blocks (5 m × 5 m). The signal strength map is shown in Figure S12. A 3D map for the RGIPT campus when a cell tower is at the original position is shown in Figure S13a. Similarly, the new proposed location of one cell tower on campus is shown in Figure S13b.

Step 7: For deploying P = 3, p is number of cell phone towers in an area. Inscribed the given Rectangle "R" in circle "O", the diameter of the circle is equal to the diagonal of "R". The Center of "R" is denoted by "C", which is now divided into three equal angles of $(360°/3) = 120°$ each. The angle will be divided into different combinations. Take the inscribed area of the rectangle and delete the other circle area. The following combinations are possible, shown in Figure S14a–c.

1.   Taking one case out of different combinations and isolating the three equal areas as shown in Figure S14, the isolated area is shown in Figure S15a.
2.   Taking one isolated area out of three mentioned as (Area 1) and fitting the area in the rectangle "LMNO" is shown in Figure S15b.

Step 8: Divide the whole isolated area into blocks of area (5 m × 5 m) within the segment divided in step 7.

Step 9: Calculate the centroid of the rectangle in which Area 1 is inscribed within the segment. The centroid is mentioned as "CArea1", as shown in Figure S16(1).

Step 10: Now, again a square is taken as in step 4 (Point 1) of dimension based on half of the distance from "C" to the centroid "CArea1". Dimension is (m × m). (* whole area will always be divided into blocks of dimension (5 m × 5 m), the only square which is taken for cell tower location correction is divided into cells of 2 m × 2 m). Similarly, as in step 5 (Point1), the square is now divided into blocks of 2 m × 2 m and repeats the same process to find the new centroid location (CArea1′), as shown in Figure S16(2).

Step 11: Determination of the average phone signal strength (each grid consists of five random points) at every 5 m × 5 m grid location from every potential tower location (xl, yl) within a segment. Where l ranges between 1 and m, m being the total number of grid cells in step 10, n is the total number of grid cells in step 8, and k is the number of generated points in n grids.

Step 12: The optimal grid cell for setting up the cellular tower within a segment is determined after comparing the phone signal strength of different grid cells for various potential tower positions within the above segment. The signal strength map of the segment is shown in Figure S16(3).

Step 13: Repeat the above steps from step 8 to step 12 for the other two segments (Area 2, Area 3). Finally, we have (CArea1′), (CArea2′), and (CArea3′) cell phone tower locations. Now, place the tower in the project area. The signal strength map for the other two segments is shown in Figure S17.

1.   Similarly, for the three locations of the tower, the signal strength map is shown in Figure S18a.

2. For the (P = 2) cell phone tower: Similarly, for the two locations of the tower, the signal strength map is shown in Figure S18b.

4.2.2. Urban Area (Shahganj Area of Agra City, India)

The urban area consists of a big population, more buildings, and difficulty in finding the best possible location for a cell phone tower. For the determination of the best location in the Shahganj area of Agra city (3 km), as shown in Figure S19a. The area itself consists of 5 Jio network towers, as shown in Figure S20a. However, the signal strength from these towers is not that accurate and adequate. There is a need for the best location for these towers.

1. Two-dimensional mapping of Agra city area: The Shahganj area of Agra city has five (JIO network towers), which are shown in Figure S20a, and the signal strength due to these towers is shown in Figure S20b. After applying the algorithm to an area of Agra city, the proposed locations for five towers have come in different sets of combinations. Out of these combinations, one has to be chosen that gives the best result. These positions are shown in Figure S21a, and the signal strength map is shown in Figure S21b. For the city location, LIDAR data are generated, which are shown in Figure S19b. For the generated LIDAR data shown in Figure S19b, DEM is created, which is shown in Figure S19c.

2. Three-dimensional mapping of Agra city area: 3D map for the previous tower at Shahganj area is shown in Figure S23a, and a 3D map for the proposed tower location is shown in Figure S23b. While determining the signal strength due to the four proposed towers, the dB m lies between −53.73 and −88.58 (dB m) and is shown in Figure S20b.

3. Effect of towers at Building: A building is taken out from the Shahganj area, and the effect of cell phone towers on that particular building is shown in Figures S21a and S24a. This is done for two cases of previously installed towers and proposed towers. It is clearly shown in Figure S24b that signal strength is improved in the case of proposed cell phone tower locations.

4. Percentage improvement in the signal strength: Analysis is performed to find the changes that occur in the signal strength map.

    a. Two cases are taken; one is when a previously installed cell phone tower is considered, and the other one is a proposed cell phone tower position. From the above Figure S24a,b, the improvement is noticeable by the signal strength legend. There is a need to calculate the average increase and decrease in signal strength. Therefore, for both cases, the map is a plot in 2D, as shown in Figure S25a,b.

    b. The two signal strength maps show that the worst value obtained is −100.68 dB m at a previously installed cell phone tower location. According to the survey, for achieving an adequate signal strength, it should be not less than −85 dB m. For the above two, it is very clear that signal strength is improved in the proposed approach.

    c. The min value of an increase in strength from case 1 (when the cell phone tower is at the original location) to case 2 (when the cell phone tower is at the proposed location) is 3.548 dB m. The max value of an increase in signal strength from case 1 to case 2 is 17.819 dB m. The average increase in the scenario is 6.845 dB m.

    d. Min value of the decrease in signal strength from case 1 to case 2 is 0.1428 dB m. Similarly, the max value of the decrease in signal strength from case 1 to case 2 is 9.415 dB m. The average decrease in the scenario is 2.157 dB m.

    e. Even the signal strength due to the proposed tower "4" instead of the proposed tower "5" is more than that of signal strength due to the pre-installed tower. It is clear from Figure S22b that instead of the proposed tower "5", tower "4" can be installed to save costs.

### 4.2.3. Non-Uniform Population Density

Determining the cell tower location for the area where there is non-uniform population density. The area of the RGIPT campus is taken, which consists of a hostel area, administrative building, and academic blocks 1 and 2 (AB1 and AB2), as shown in Figure 13a. The cell tower location is calculated according to the method discussed above. The signal strength map is shown in Figure 13b.

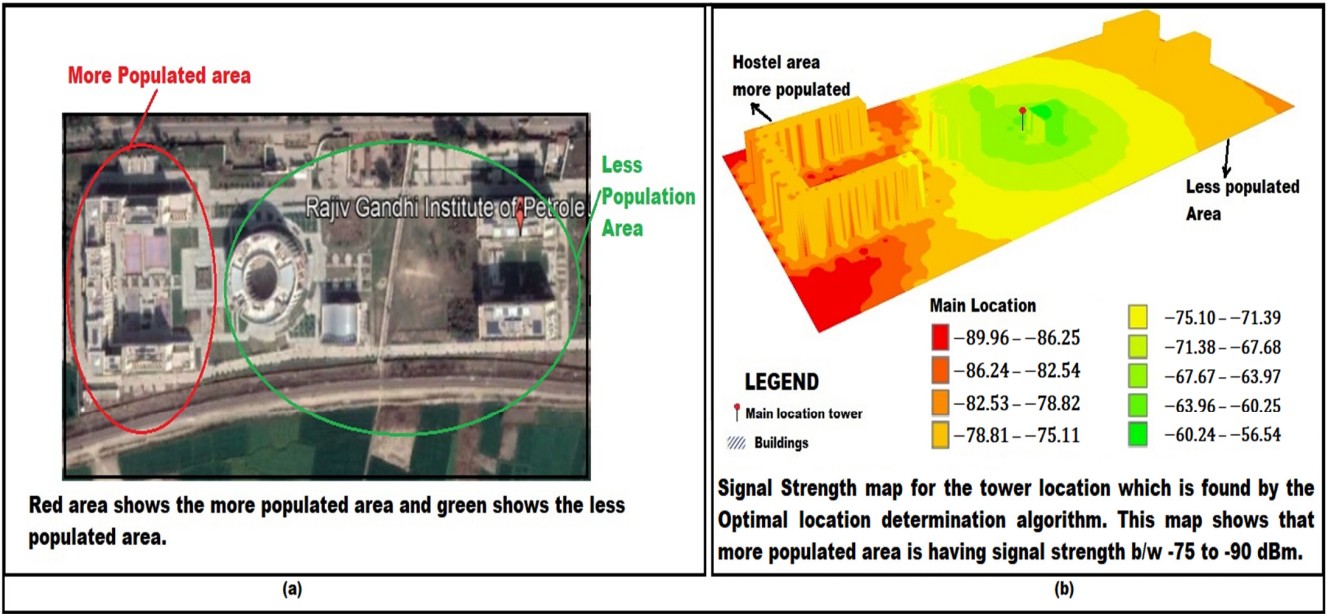

**Figure 13.** (**a**) RGIPT area with non-uniform population. (**b**) Signal strength map for the main location initially determined by the proposed method.

The population in the hostel area is more compared to the rest of the area. Sometimes there is a need for better signal strength in the highly populated area as most of the population is there. Initially, the cell tower location using the proposed method is determined, which is irrespective of the population density. The signal strength in Figure 13b shows that both the high and low-populated areas have moderate signal strengths. Therefore, the area from the cell tower location in Figure 13b is divided into two parts according to the area divided (here divided in two halves as high and low population). To determine the cell tower location for the two halves, the above procedure is used. Signal strength for both halves is shown in Figure 14a,b.

According to the proposed method, which is discussed above, the whole area in Figure 13a is divided into 6050 equal cells of 5 m × 5 m, where 1800 cells are from a highly populated area, and 4250 cells are from a lowly populated area. Changes in signal strength concerning cells for the two halves are shown in Figure 15a. The total number of cells lies in (range −50 to −75 dB m) and in (range −75 to −95 dB m) for the main tower location, locations "1" and "2" are shown in Figure 15b. Percentage change in signal strength for locations "1" and "2" to the main location of the cell phone tower is mentioned in Table 1.

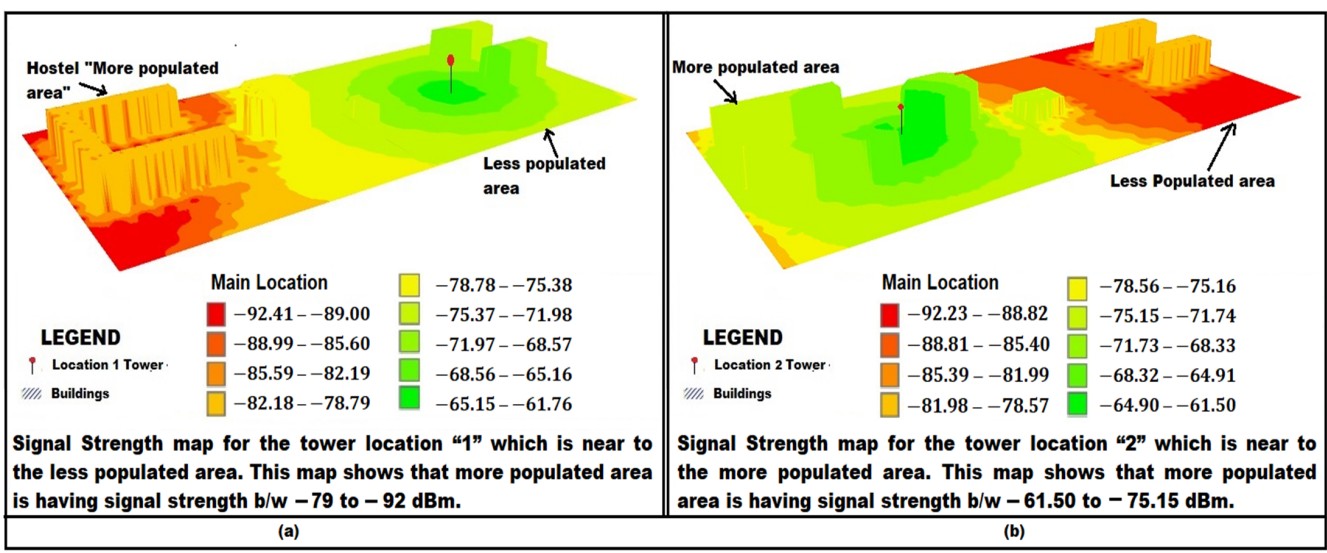

**Figure 14.** (**a**) Signal strength map for cell tower location "1". (**b**) Signal strength map for cell tower location "2".

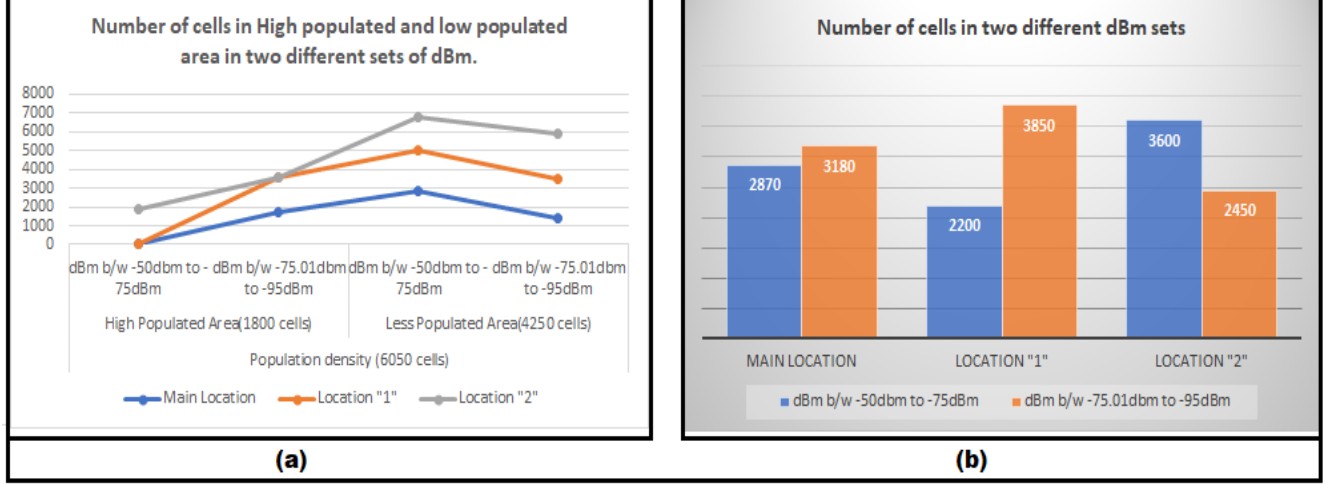

**Figure 15.** (**a**) Number of cells between the high and low-populated areas according to the two dB m sets for the three locations of the tower. (**b**) Number of cells for the whole area for the three locations of the tower.

**Table 1.** Percentage change in locations "1" and "2" from cell values of the main location.

| | Improvement in Signal Strength (RGIPT Small Area Total Cell 6050 Cells of (5 m × 5 m) Grid) |
|---|---|
| | % Population gets (range −50 to −75 dB m) |
| Main Location | 47.43801653 |
| Location "1" (Tower towards lower population) | 36.36363636 |
| Location "2" (Tower towards higher population) | 59.50413223 |

## 5. Conclusions

The analysis comprises determining the pathways of signal propagation from the cellular tower location. The signal strength is analyzed at each cell phone user location (receiver's position). The determined propagation routes of the signal follow the shortest route, thus improving accuracy in the determination of signal strength. The signal strength

determination model is developed based on a 3D rigorous novel point-to-point routing methodology. The signal strength determination model is designed using semi-empirical models of propagation. It is validated with the data of APP-recorded ground-based signal strengths of over 1000 points at the RGIPT and Agra sites. Using the least square technique, the determined model gave an RMSE error of 3.45. The technique performed well and can adapt with 3D LiDAR data, unlike well-known algorithms that are conventionally used for finding the shortest route in 2D. The use of 2D data and constraints of data inputs prevent accurate prediction of signal strength. For example, different parts of the RGIPT campus are predicted to have uniform signal strength, whereas detailed variations in signal strength are modeled for the site. The ground observations corroborated the modeled predictions. Further, the 2D data never allow for any prediction in 3D, e.g., the different floors of a building can have different qualities of signal strength. Therefore, to have a precise determination, high-quality 3D terrain data is required. Thus, the 3D Lidar-based prediction methodology very significantly improved the prediction resolution and accuracy.

The approach described above aids in the placement of cell phone towers to ensure improved signal strength at each cell phone user location. The most appropriate height (Z) and position (X, Y) of a cellular tower for the project area of the RGIPT campus is determined to achieve a signal strength between −50 and −90 dB m using the model established above, as shown in Figure S16. Further, users' settlement density can also be considered to customize the cell phone tower location to provide a very high quality between −50 and −75 dB m or better signal to the majority of the campus users. After keeping the population in mind, 22% of the population who initially got a signal strength between −75 and −95 dB m now receives a signal strength above −75 dB m. The model is applied to determine the cellular tower distribution for the Shahganj area of Agra city, where five towers were previously installed. The five modeled cell phone tower locations indicated, if installed at modeled locations, could have very significantly improved the signal strength quality from −50 to −75 dB m, as shown in Figures S18–S20. Further, the model has indicated that the Shahganj area could have managed the existing quality of signal strength using only four towers instead of five. The model is developed in such a way that it will find out the optimal location for the cell phone tower and evaluate the signal strength at each user location. The centroid-based approach and rough to fine-scale optimization (100 m × 100 m to 2 m × 2 m) offered precision without restricting very high time constraints.

## 6. Future Scope

The developed model offers a unique opportunity to predict the signal strength at a place three-dimensionally and accurately. Further, it can optimally design the number and location(s) for setting up the new cellular tower(s) to offer improved quality of signal strength. The research automatically foresees the opportunity to make two web APPs, one for the users and another for the cellular service providers. Users can choose the best cellular service provider for a place (in terms of signal quality), and the service provider can use it to determine where to set up their cellular tower, how high it should be, how many towers are required to be placed, and what will be the quality of signal strength at a particular user location.

**Supplementary Materials:** The following supporting information can be downloaded at: https://www.mdpi.com/article/10.3390/asi5020030/s1, Figure S1: (**a**) Generated LIDAR data for the RGIPT campus area, (**b**) Digital Elevation Model for the RGIPT area; Figure S2: The small area of the RGIPT campus for cell phone tower height determination; Figure S3: (**a**) Signal strength map when Z is at 0 m (**b**) Signal strength map when Z is at 5 m height; Figure S4: (**a**) Signal strength map when Z is at 15 m (**b**) Signal strength map when Z is at 25 m height; Figure S5: (**a**) Signal strength map when Z is at 30 m (**b**) Signal strength map when Z is at 35 m height; Figure S6: (**a**) MATLAB plot for the project Area of RGIPT, (**b**) Building edges and corner estimation on the RGIPT campus; Figure S7: (**a**) Rectangle ABCD into several blocks of 5 m × 5 m dimensions, (**b**) Random generation of points in each block; Figure S8: Centroid is taken initially for processing; Figure S9: Different shapes taken for

choosing the dimension of buffer near the centroid region; Figure S10: Signal strength for the L-shape to choose the best location with a chosen buffer region; Figure S11: (**a**) Another L-shaped case is taken to approximate the dimension of buffer, (**b**) U-shaped case for buffer dimension approximation; Figure S12: Signal strength map for one proposed cell phone tower on campus; Figure S13: (**a**) Signal strength 3D map for the RGIPT campus from Jio tower placed outside campus, (**b**) Signal strength 3D map for the RGIPT campus from one proposed Jio tower inside campus; Figure S14: Combinations of angle area division into three equal parts; Figure S15: (**a**) Three equal areas (Area 1, Area 2, and Area 3) of the chosen combination of angle division, (**b**) Area 1 of RGIPT is inscribed in a rectangle; Figure S16: Buffer region selection for correction of the centroid of Area 1; Figure S17: Signal strength map for Areas 2 and 3; Figure S18: (**a**) Signal strength map for three proposed cell phone towers on campus, (**b**) Signal strength map for two proposed cell phone towers on campus; Figure S19: (**a**) Shahganj area of Agra city (Urban area) chosen for the determination of the best location of cell phone tower, (**b**) Shahganj area LIDAR generated data, (**c**) Digital Elevation Model for the Shahganj area of Agra city; Figure S20: (**a**) Previously installed cell phone tower location. (**b**) Signal strength due to the five cell phone towers; Figure S21: (**a**) Five proposed cell phone tower locations. (**b**) Signal strength due to these five proposed cell phone towers; Figure S22: (**a**) Four proposed cell phone tower locations. (**b**) Signal strength due to these five proposed cell phone towers; Figure S23: (**a**) Three-dimensional signal strength map for the Shahganj area due to previously installed cell phone tower locations, (**b**) 3D signal strength map for the Shahganj area due to proposed cell phone tower locations; Figure S24: (**a**) Effect of previously installed cell phone towers on building walls and corners, (**b**) Effect of proposed cell phone towers on building walls and corners; Figure S25: (**a**) Optimization map for previously installed cell phone towers. (**b**) Optimization map for proposed cell phone towers.

**Author Contributions:** Conceptualization, S.B. (Shruti Bharadwaj) and S.K.T.; Data curation, M.I.Z. and R.D.; Formal analysis, S.B. (Shruti Bharadwaj); Funding acquisition, M.I.Z.; Investigation, M.I.Z.; Methodology, S.B. (Shruti Bharadwaj) and R.D.; Project administration, S.B. (Shruti Bharadwaj); Software, S.K.T. and R.D.; Supervision, S.K.T. and S.B. (Susham Biswas); Validation, S.K.T.; Visualization, S.B. (Susham Biswas) and R.A.F.; Writing—original draft, S.B. (Shruti Bharadwaj), R.D. and R.A.F.; Writing—review and editing, S.B. (Susham Biswas). All authors have read and agreed to the published version of the manuscript.

**Funding:** This research received no external funding.

**Data Availability Statement:** Data utilized in this research work can be downloaded from Google Earth.

**Acknowledgments:** The authors are thankful to Rajiv Gandhi Institute of Petroleum Technology, A.S.K. Sinha, for instilling a positive research environment on campus. Also, the authors are thankful to Rashid Aziz Faridi, Department of Geography at Aligarh Muslim University for his support in writing this article.

**Conflicts of Interest:** The authors declare no conflict of interest.

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
