# Peer review of "A Novel Method to Determine the Optimal Location for a Cellular Tower by Using LiDAR Data"

_asi, doi:10.3390/asi5020030_

Round 1
Reviewer 1 Report
Paper is well written and contains sound results. I have following suggestions for improvement;
Proof read paper for English language and grammar.
Use uniform formatting for equations throughout text (For example see page 12 to 13)
Add 1 to 2 latest references if possible.
Author Response
Reviewer 1:
The Paper is well written and contains sound results. I have the following suggestions for improvement;
Proof read paper for English language and grammar.
The authors have read the manuscript again and have omitted the grammatical errors.
Use uniform formatting for equations throughout text (For example see page 12 to 13)
The authors thank the reviewer for pointing out this inconsistency in formatting and the text has been made uniform throughout the manuscript.
Add 1 to 2 latest references if possible.
The authors have added some latest references.
Reviewer 2 Report
The paper deals with novel method to determine the optimal location for a cellular tower by using LiDAR data. The optimization of placement of cellular towers is an interesting issue and an area that currently needs to be addressed in connection with the massive use of mobile networks. In this case, I consider it very important to address this issue and this article will certainly be interesting for readers of this magazine.
The article contains a very detailed description of the problem and other associated problems with relevant references. The authors also describe similar works and describe other authors' methods of solving similar problems.
The goal of the suggested research is to figure out where the best cell phone tower should be placed. The rapid increase in the use of mobile technologies is creating an increased demand for signal coverage quality, so suitable cellular tower placement is a key task in ensuring signal coverage.
Determination of optimal location of cellular tower is associated with many relevant problems related to geographical topology, placements of the building’s features, terrain data etc.
A new methodology for solving of defined problem is proposed. Cell phone signal strength determination model is presented as flowchart. The key problem for this methodology is determination of terrain parameters between the cellular tower and user position. LIDAR terrain determination is proposed to use. Also model for determination of optimal location for setting up a cell phone tower is the main contribution of this paper.
The proposed method is applied to RGIPT campus, what is urban city area. The optimal location for a cellular tower will guarantee that the signal strength is sufficient for all nearby cell phone users.
The signal strength from the nearest cell phone tower was determined experimentally and recorded by a mobile application called (Network cell info lite) shown in the article. Then signal strength map was created from observed data.
The authors assume that there are some analogies between signal transmission losses and sound transmission losses. This assumption is correct for creating a methodology for finding the optimal position of the cellular tower.
The article is very valuable but the authors have made some serious mistakes that need to be corrected.
Comments:
1.
To clarify the situation, FIG. 8 needs to be supplemented with an better indication of the location of the nearest cellular towers. Or please add another picture where it will be clearly displayed.
There is a only Jio tower indication – please use any more visible symbol. But it would be good to show the situation how far the other towers are if possible to illustrate the overall situation, because the reader does not know it.
2.
Neither equation in the article is numbered. All equations are lost in the text. It is very confusing and a scientific article must not look like this. Please number all equations and use a uniform style for writing equations. Consistently use variables with a description of the meanings of individual symbols after the equations. And then in the text you can refer to the individual equations by numbering.
Often the equations are written as text only. For example: “Distance Attenuation= 20 LOG10(D)+11”
Please use the equation editor. Then the variables will be like italic and the matrices will be like bold style. Please, also use the same style for writing of variables inside text.
Do not use variables marked with two letters, for example you use the variable "SR" and this may look like a scalar product of two variables. Please use subscripts.
3.
Please check the numeric values in the article. Some are listed without units even though they have their own unit. Please be consistent.
4.
Finally, please complete your other plans for the future as you plan to continue this research.
5.
Reference 22 is not complete (volume, issue). There are missing all information about this reference. Also Reference 31 has no volume number and issue. Reference 34 is also not completed.
Author Response
Reviewer 2:
- To clarify the situation, FIG. 8 needs to be supplemented with an better indication of the location of the nearest cellular towers. Or please add another picture where it will be clearly displayed.
There is a only Jio tower indication – please use any more visible symbol. But it would be good to show the situation how far the other towers are if possible to illustrate the overall situation, because the reader does not know it.
The authors thank the reviewer for their comments. In the revised version, a new figure (Figure 7 in the revised version) is added to properly describe the location of the nearest cellular towers.
In the figure 8 of the revised manuscript, appropriate changes have been made in the symbols to make them look more visible. In the figure 7, it can now be clearly seen how far the other towers are.
- Neither equation in the article is numbered. All equations are lost in the text. It is very confusing, and a scientific article must not look like this. Please number all equations and use a uniform style for writing equations. Consistently use variables with a description of the meanings of individual symbols after the equations. And then in the text you can refer to the individual equations by numbering.
Often the equations are written as text only. For example: “Distance Attenuation= 20 LOG10(D)+11”
Please use the equation editor. Then the variables will be like italic and the matrices will be like bold style. Please, also use the same style for writing of variables inside text.
Do not use variables marked with two letters, for example you use the variable "SR" and this may look like a scalar product of two variables. Please use subscripts.
The authors understand the reviewer’s concern and regret the mentioned fallacies. The authors have numbered all the equations, have used a consistent style of writing. In the revised manuscript, the meaning of individual symbols has been mentioned, and the symbols used throughout the manuscript is consistent. Also, the distance attenuation equation has been written like an equation, instead of a text.
The authors have used the equation editor, the variables have been mentioned in italics and the matrices in bold. The authors have ensured a consistency in writing the variables inside the text.
The authors have made sure to include subscripts, to avoid any confusion.
- Please check the numeric values in the article. Some are listed without units even though they have their own unit. Please be consistent.
The authors have mentioned the units for all those numeric values possessing a unit.
- Finally, please complete your other plans for the future as you plan to continue this research.
This manuscript is more relevant to developing countries than developed countries, especially India. In India, setting up a tower is costly, so the authors wanted to show in this research that there are several cases where the cellular companies set up the towers without knowing the optimal location where they should be installed. In some cases, this leads to overdesigning while in others, there is under designing. The developed model offers a unique opportunity to predict the signal strength at a place in 3D and accurately. Further, it can optimally design the number and location(s) for setting up the new cellular tower(s) for a place to offer improved quality of signal strength. The research automatically foresees the opportunity to make two web APPs, one for the users and another for the cellular service providers. Users can choose the best cellular service provider for a place (in terms of signal quality) and the service provider can use it to determine where to set up their cellular tower, how high it should be, how many such towers are required to be placed, and what will be the quality of signal strength at a particular user location, etc. Currently, this research has been performed on a laboratory scale, and with collaborations with industrial partners, the authors wish to scale up this work across India and even across other countries. The authors have also mentioned their plans for the future in the manuscript.
- Reference 22 is not complete (volume, issue). There are missing all information about this reference. Also, Reference 31 has no volume number and issue. Reference 34 is also not completed
The authors thank the reviewer for their observation. The references have been cross-checked and all the errors have been omitted in the revised version.
Reviewer 3 Report
I do believe the authors did quite a lot of work, but I am overwhelmed by the massive information provided in the lengthy text. The topic is interesting and useful, but I cannot see the innovation. I'm not sure what unique role lidar data plays, and what the "Novel Method" refers to. This article appears to be a report of an internship project rather than a scientific paper. The author should first learn how to organise an academic paper, how to state the contribution and conclusions of the work, rather than listing what has been done. The figures and tables in the manuscript are unprofessional and need further improvement.
Author Response
Reviewer 3:
I do believe the authors did quite a lot of work, but I am overwhelmed by the massive information provided in the lengthy text. The topic is interesting and useful, but I cannot see the innovation. I'm not sure what unique role lidar data plays, and what the "Novel Method" refers to. This article appears to be a report of an internship project rather than a scientific paper. The author should first learn how to organize an academic paper, how to state the contribution and conclusions of the work, rather than listing what has been done. The figures and tables in the manuscript are unprofessional and need further improvement.
The existing optimum cellular tower locating algorithms are primarily theoretical or are using very simplified rough 2D terrain data of land-use and landcover to ascertain the best location for setting up a cellular tower using simplified objective functions. In absence of a detailed and practical approach to adapt to any urban setup precisely, the GIS approaches follow average techniques to determine the optimality. It invariably results over or under design. There is thus a need to evolve a technique that should be able to use the detailed variation in topography for an area and estimate the detailed obstructions or impacts of each for the signals to reach any user location from a cell phone tower. It is required to be understood that the urban environment having adjacent high-rise buildings located between the cellular tower and user location play a very big role in terms of obstruction of signals. Thus, 2D data and rough land use and landcover data significantly preclude estimation of detail and accurate signal strength for a place. Developing countries have a large proportion of unplanned cities offering a very mixed land use and land cover and often create a significant challenge in terms of accurate estimation of signal strength for a place from a cellular service provider. It is thus required to come up with a technique that can model or compute accurate signal strength for a place for a user about a given location for a cell phone tower. The model should also be able to estimate the best location for setting up a cellular tower for a place or estimate numbers and locations of possible cellular towers for an area to provide good signal strength to every user location. Existing GIS software suffers in terms of its ability to accommodate huge terrain data or extract building and other obstructions in 3D automatically which prevent direct transmission of a signal from the cellular tower(s) to different user locations. There is thus a requirement to use 3D terrain data of high precision, which can find out a technique to extract all obstructions between any pair of the cellular tower and user locations, determine losses in transmission at its every route of transmission and can estimate an optimization algorithm which can work with all the spatial attributes (transmission paths, distances, transmission losses, etc.) and non-spatial attributes (number of users at a building, population density, etc.) to determine the signal strength and optimum location(s) for the setting-up cellular tower(s). LiDAR data offers very precise 3D data in point cloud form. It is required to establish a novel cellular tower locating optimization technique using LiDAR data which will overcome challenges of accommodating precise 3D data and enable highly accurate optimization, in terms of offering the best location(s) for the setting-up cellular tower(s) and giving the best estimate of signal strength for any surrounding areas.
The authors have improved the organization of the article, and the figures and tables have been vastly improved. The readability of the manuscript has been improved. The authors are hopeful that the reviewer will be happy with the revised version.
Round 2
Reviewer 3 Report
In Figure 4,5,6, the black background is not professional.
The subheadings in Section 4 should be reconsidered to better demonstrate the logic of the manuscript.
The non-essential content of Section 4 may be considered for inclusion in the additional material to streamline the main body of the manuscript.
The authors should provide a pure revised version without change tracks.
Author Response
- In Figure 4,5,6, the black background is not professional.
The authors thank the reviewer for their observation. In the Figure 4,5,6 of the revised version, appropriate changes have been made to make them look more visible.
- The subheadings in Section 4 should be reconsidered to better demonstrate the logic of the manuscript.
The subheading of the section 4 of the manuscript have been attempted to be modified by the authors
- The non-essential content of Section 4 may be considered for inclusion in the additional material to streamline the main body of the manuscript.
The authors have improved the organization of the article, and the figures and content from subsections 4.1.2, 4.2.1, 4.2.2 have been included in the supplementary file. The readability of the manuscript has been improved. The authors are hopeful that the reviewer will be happy with the revised version.
- The authors should provide a pure revised version without changing tracks.
The authors are now submitting the manuscript without changing tracks.